# Megakaryocyte- and erythroblast-specific cell-free DNA patterns in plasma and platelets reflect thrombopoiesis and erythropoiesis levels

Joshua Moss [1,2,6], Roni Ben-Ami [1,6], Ela Shai[3], Ofer Gal-Rosenberg [1], Yosef Kalish[3], Agnes Klochendler[1], Gordon Cann[4], Benjamin Glaser [5], Ariela Arad[3] ✉, Ruth Shemer[1] ✉ & Yuval Dor [1] ✉

Circulating cell-free DNA (cfDNA) fragments are a biological analyte with extensive utility in diagnostic medicine. Understanding the source of cfDNA and mechanisms of release is crucial for designing and interpreting cfDNA-based liquid biopsy assays. Using cell type-specific methylation markers as well as genome-wide methylation analysis, we determine that megakaryocytes, the precursors of anuclear platelets, are major contributors to cfDNA (~26%), while erythroblasts contribute 1–4% of cfDNA in healthy individuals. Surprisingly, we discover that platelets contain genomic DNA fragments originating in megakaryocytes, contrary to the general understanding that platelets lack genomic DNA. Megakaryocyte-derived cfDNA is increased in pathologies involving increased platelet production (Essential Thrombocythemia, Idiopathic Thrombocytopenic Purpura) and decreased upon reduced platelet production due to chemotherapy-induced bone marrow suppression. Similarly, erythroblast cfDNA is reflective of erythrocyte production and is elevated in patients with thalassemia. Megakaryocyte- and erythroblast-specific DNA methylation patterns can thus serve as biomarkers for pathologies involving increased or decreased thrombopoiesis and erythropoiesis, which can aid in determining the etiology of aberrant levels of erythrocytes and platelets.

Circulating cell-free DNA (cfDNA) molecules are thought to be released from dying cells and can be used to monitor tissue turnover rates in health and disease. Fetal DNA present in maternal circulation allows for detection of fetal aneuploidies[1]; donor-derived DNA in the circulation of organ transplant recipients provides a non-invasive marker of graft rejection[2]; and tumor-derived mutant cfDNA allows for detection and monitoring of cancer[3].

Accurate identification of the tissue origins of cfDNA holds great potential for sensitive monitoring of turnover dynamics in specific tissues and cell types, in health and disease. DNA methylation patterns

[1]Department of Developmental Biology and Cancer Research, Institute for Medical Research Israel-Canada, the Hebrew University-Hadassah Medical School, Jerusalem, Israel. [2]Sharett Institute of Oncology, Hadassah-Hebrew University Medical Center, Jerusalem, Israel. [3]Hematology Department, Hadassah-Hebrew University Medical Center, Jerusalem, Israel. [4]GRAIL, LLC., Menlo Park, CA, USA. [5]Endocrinology and Metabolism Service, Hadassah University Medical Center and Faculty of Medicine, the Hebrew University, Jerusalem, Israel. [6]These authors contributed equally: Joshua Moss, Roni Ben-Ami. ✉e-mail: arielaar@hadassah.org.il; shemer.ru@mail.huji.ac.il; yuvald@ekmd.huji.ac.il

provide means for determining the tissue origins of cfDNA, given the cell type specificity of this epigenetic mark[4]. Indeed, we and others have shown that tissue-specific methylation patterns can serve as cfDNA biomarkers for elevated turnover in specific tissues. For example, cardiomyocyte DNA is present in the plasma of patients following myocardial infarction[5]; hepatocyte cfDNA is present in the plasma of patients with liver damage[6], and exocrine pancreas DNA is found in the plasma of patients with pancreatitis[7]. Additionally, novel assays detect cancer-related methylation aberrations for early diagnosis of cancer[8]. Alternative approaches for determining the tissue origins of cfDNA rely on tissue-specific patterns of nucleosome positioning[9], fragmentation, topology and size of cfDNA[10].

More recently, patterns of histone modification in circulating chromatin fragments were used to infer gene expression and hence identity of the cells that give rise to cfDNA[11]. Such analyses of the circulating epigenome demonstrate that plasma cfDNA is mostly derived from cells of hematopoietic origin, specifically neutrophils (30%), lymphocytes (12%) and monocytes (11%), as well as vascular endothelial cells (~10%) and hepatocytes (1–3%)[12]. However, there have been contrasting reports regarding the contribution of cfDNA from erythroblasts and megakaryocytes, cells which arise from two related but distinct blood lineages. Lam et al. identified genomic loci that are unmethylated specifically in erythroid cells and assessed their level in cfDNA. They concluded that ~30% of cfDNA originates from erythroid cells, and that anemia and thalassemia impact the level of erythroid cfDNA[13]. Consistent with this, our own deconvolution of the plasma methylome suggested that erythrocyte progenitors contribute to ~30% of cfDNA in healthy individuals[12]. However, these studies did not take into account the relative contribution of megakaryocytes to cfDNA. Sadeh et al. concluded from immunoprecipitation of circulating chromatin that megakaryocytes, rather than erythroblasts, are major contributors of nucleosomes to healthy plasma[11].

In this study, we identify definitive methylation markers that distinguish megakaryocytes and erythroblasts and use these to characterize the presence of DNA from megakaryocytes and erythroblasts in plasma of healthy individuals and in different pathological scenarios which affect the production of platelets and red blood cells. We find that measuring megakaryocyte and erythroblast cfDNA allows the detection and distinction of pathologies affecting these lineages, even when not reflected in peripheral blood cell counts. Furthermore, we discover that platelets, not previously believed to contain genomic DNA, do in fact contain genomic DNA originating in megakaryocytes, despite the lack of a nucleus.

## Results

### Methylomes of erythroblasts and megakaryocytes and their representation in healthy plasma

Due to previous reports that non-leukocyte cells of hematopoietic origin are major contributors to cfDNA, we sought to evaluate the relative contribution of erythroid and megakaryocyte genomes. To this end, we isolated DNA from bone-marrow erythroblasts (Supplementary Fig. S1) as well as multiple types of white blood cells (see methods) and performed whole-genome bisulfite sequencing to obtain a genome-wide methylation profile of each cell type. We also obtained previously published genome-wide methylation profiles of megakaryocytes as well as common megakaryocyte-erythroid progenitors for comparison. This comparison revealed multiple loci that were uniquely methylated or unmethylated in erythroblasts (1884 sites) and in megakaryocytes (96 sites) and could in principle serve as specific biomarkers for DNA derived from these cell types (Fig. 1A, B). The erythroblast genome was largely unmethylated, in line with previous reports describing gradual genome-wide demethylation of erythroid cells in mice and humans during their terminal differentiation[14,15] (Supplementary Fig. S2).

In order to evaluate the contribution of these genomes to cfDNA, we obtained genome-wide methylation profiles of purified white blood cells (WBC) and cfDNA from 23 healthy individuals, sequenced at ~85x coverage[16]. Methylation levels of genomic regions uniquely unmethylated in megakaryocytes were ~26% less methylated in cfDNA compared to WBC-derived DNA (Fig. 1A), suggesting that megakaryocyte genomes are major contributors to cfDNA. In contrast, genomic regions uniquely unmethylated in erythroblasts were only ~4% less methylated in cfDNA compared to WBC-derived DNA (Fig. 1B), suggesting that erythroblast genomes are minor contributors to cfDNA in healthy individuals. The fact that erythroblast and megakaryocyte markers are methylated in WBC further supports the idea that they originate in cells that are not present in circulation. Targeted bisulfite PCR-sequencing of regions uniquely unmethylated in megakaryocytes or erythroblasts in cfDNA or blood was consistent with the genome-wide analyses, demonstrating elevated levels of megakaryocyte as well as erythroblast DNA in plasma as compared to whole blood (Fig. 1C, D, Supplementary Fig. S3, S4).

### Platelets contain genomic DNA from MK

Platelets are products of megakaryocytes, that contain megakaryocyte-derived pre-mRNA and mRNA[17] and carry out splicing and protein synthesis, but do not contain a nucleus and are not thought to contain genomic DNA. However it has been reported that platelets contain histone proteins[18], raising the possibility that they may carry some megakaryocyte DNA as well. To test this idea, we isolated platelets (see methods) and performed DNA extraction (Supplementary Table S2). We reasoned that DNA found in platelets could be genomic DNA of a megakaryocyte that was trapped within forming platelets, or alternatively could be cfDNA from other cell types that adhered to the external surface of platelets or was internalized, similar to tumor-derived mRNA which has been suggested to be present in platelets of individuals with cancer[19,20]. To distinguish between these possibilities, we performed targeted bisulfite sequencing of regions uniquely unmethylated in megakaryocytes in the platelet DNA concentrates. Strikingly, platelet DNA was found to be mostly unmethylated at these loci (~80%), suggesting megakaryocytes as the main origin of platelet DNA (Fig. 2A). Platelet concentrates also contained DNA methylation markers of leukocytes and hepatocytes, suggesting the presence of some DNA from these cell types (Fig. 2A). To distinguish between DNA associated with the external surface of platelets and DNA present within platelets we treated platelet isolates with DNaseI, reasoning that this enzyme would have an effect only on external DNA. Strikingly, DNase treatment did not reduce the concentration of megakaryocyte DNA in platelets but did eliminate leukocyte- and hepatocyte-derived DNA (Fig. 2B and Supplementary Fig. S5), suggesting that the megakaryocyte DNA is present within the platelet. Interestingly, platelet DNA is present in the form of large molecular weight DNA, in contrast to nucleosome-size fragments of typical cfDNA (Supplementary Fig. S6). Notably, the total amount of DNA present in platelets (~$2 \times 10^{-6}$ genomes/platelet) suggests that only a small fraction (~0.1%) of a megakaryocyte genome DNA is present in the platelets that derive from this cell (Supplementary Table S2).

### Genome-wide analysis of platelet DNA

In order to confirm that platelet DNA is megakaryocyte-derived, we performed whole-genome bisulfite sequencing of platelet DNA ($n = 3$ individuals). Platelets contained DNA from all human chromosomes, with an enrichment for mitochondrial DNA (~59% of DNA content). This is consistent with the reported presence of ~5 mitochondria per platelet[21], and suggests that each platelet contains ~56 kb of genomic DNA, in accordance with our estimate that ~0.1% of a megakaryocyte genome is present in platelets originating from one megakaryocyte (given a mean megakaryocyte ploidy of 12N[22] and ~1000–4000 platelets produced per megakaryocyte[23]) (Fig. 3A). Interestingly, the relative frequency of autosomal chromosomes in platelets was not random,

with overrepresentation of certain chromosomes (ex. chromosome 4) and underrepresentation of others (ex. chromosome 17) (Supplementary Fig. S7). We then focused on regions of the genome uniquely unmethylated in different hematopoietic cell types as well as other cell types which contribute to cfDNA such as endothelial cells and hepatocytes, and evaluated the methylation of these regions in platelet DNA. Importantly, only the megakaryocyte-unmethylated regions were unmethylated in platelets (Fig. 3B, C, Supplementary Fig. S8). Megakaryocyte-unmethylated regions were shared across different ploidy levels (Supplementary Fig. S9). Deconvolution analysis of the DNA methylation profile of platelet DNA further identified megakaryocytes as the major contributors to platelet DNA (Fig. 3D).

### The origins of MK-derived cfDNA

After establishing that megakaryocyte-derived DNA is present in plasma, as well as in platelets, we aimed to determine whether megakaryocyte-derived cfDNA originates directly from megakaryocytes, or alternatively from platelets (either whole platelets present in cfDNA preparations, or platelets that release cfDNA). To this end, we obtained plasma samples from females who had received a transfusion of platelets from male donors ($n = 5$). We reasoned that if megakaryocyte cfDNA is derived from platelets, the cfDNA of these female recipients should contain a large proportion of DNA containing the Y-chromosome. Interestingly, Y-chromosome derived DNA could not be detected in these plasma samples more than 24 h post-transfusion, even though the half-life of transfused platelets is generally >2 days[24], and even when male DNA is observed in platelets (Fig. 4, Supplementary Figs. S10 and S11). These findings suggest that megakaryocyte plasma cfDNA is likely not derived from DNA found in platelets, but rather directly from megakaryocytes, consistent with copy number changes that are present in platelet DNA but not in either megakaryocyte DNA or cfDNA (Supplementary Fig. S7).

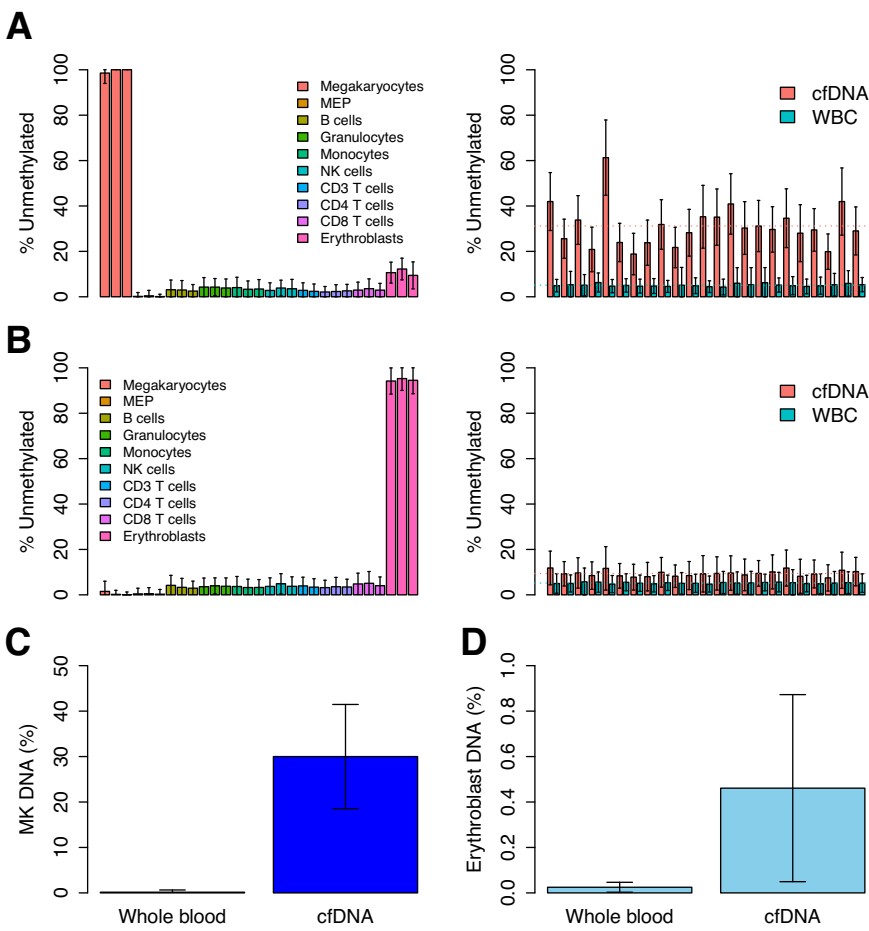

**Fig. 1 | Comparative analysis of erythroblast and megakaryocyte methylomes, and their representation in plasma. A** Genomic loci unmethylated specifically in megakaryocytes ($n = 96$ loci) were identified by comparison to other cell types of hematopoietic origin (left) and subsequently evaluated by WGBS of white blood cells (WBC) and cell-free DNA (cfDNA) of 23 individuals (right). These sites were, on average, 5% unmethylated in WBC and 31% unmethylated in cfDNA. For each individual, the sites were significantly hypomethylated in cfDNA ($p < 7.3$e-17, paired two-tailed Mann-Whitney test). Average percent of unmethylated sites is marked by a dotted line for cfDNA (red) and WBC (blue). Error bars represent standard deviation. **B** Genomic loci unmethylated specifically in erythroblasts ($n = 1884$ loci) were identified by comparison to other cell types of hematopoietic origin (left) and subsequently evaluated by WGBS of white blood cells (WBC) and cell-free DNA (cfDNA) of 23 individuals (right). These sites were, on average, 5% unmethylated in WBC and 9% unmethylated in cfDNA. For each individual, the sites were significantly hypomethylated in cfDNA ($p < 3.7$e-67, two-tailed Mann-Whitney test). Average percent of unmethylated sites is marked by a dotted line for cfDNA (red) and WBC (blue). Error bars represent standard deviation. **C** Targeted bisulfite-sequencing of megakaryocyte-specific unmethylated regions in whole blood ($n = 26$) and cfDNA ($n = 62$) of healthy individuals, demonstrating a higher percentage of megakaryocyte-derived DNA in cfDNA compared to WBC ($p = 1.7$e-13, two-tailed Mann-Whitney test). Error bars represent standard deviation. **D** Targeted bisulfite-sequencing of erythroblast-specific unmethylated regions in whole blood ($n = 15$) and cfDNA ($n = 71$) of healthy individuals, demonstrating a higher percentage of erythroblast-derived DNA in cfDNA compared to WBC ($p = 1.4$e-8, two-tailed Mann-Whitney test). Error bars represent standard deviation. Source data are provided as a Source Data file.

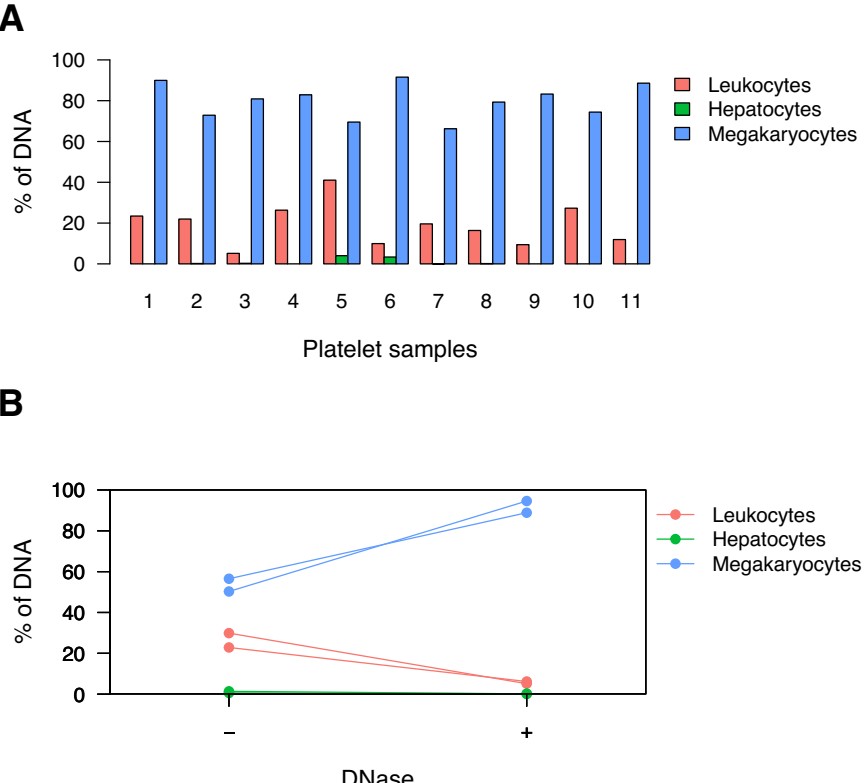

Fig. 2 | **Platelets contain genomic DNA derived from megakaryocytes. A** DNA extracted from platelet concentrates contains DNA derived from megakaryocytes, leukocytes and hepatocytes, as measured by targeted bisulfite sequencing of cell-type specific unmethylated genomic regions. **B** Uncentrifuged platelet concentrates (platelets in plasma) were analyzed with or without DNase treatment, demonstrating that DNase treatment reduces leukocyte and liver, but not MK markers in platelets. Source data are provided as a Source Data file.

## A survey of megakaryocyte and erythroblast cfDNA in relevant diseases

After establishing that circulating megakaryocyte DNA originates from megakaryocytes and not from platelets, we reasoned that its concentration in plasma may be reflective of megakaryocyte activity and turnover and hypothesized that megakaryocyte cfDNA would increase in cases of hyperproliferation of megakaryocytes and decrease in cases of hypoproliferation. To test this hypothesis, we collected samples from healthy individuals ($n = 77$), from patients with low platelet counts associated with hypoproliferation of megakaryocytes due to chemotherapy treatment ($n = 5$) and from patients with low platelet counts associated with hyperproliferative megakaryocytes due to peripheral destruction of platelets (Immune Thrombocytopenia [ITP]) ($n = 5$). We also collected plasma from patients with high platelet counts due to hyperproliferation of megakaryocytes (Essential Thrombocythemia [ET]) ($n = 5$). Healthy individuals had on average 510 genome equivalents of megakaryocyte cfDNA/ml plasma (GE/ml) (Table 1, Supplementary Table S3). ET patients had significantly elevated levels of megakaryocyte cfDNA (average 4242 GE/ml, $p < 0.05$). Interestingly, while patients with a hypoproliferative bone marrow and patients with ITP had similar platelets counts, ITP patients had elevated concentrations of circulating megakaryocyte DNA (average 1608 GE/ml, $p < 0.05$) while patients with hypoproliferative bone marrows had reduced concentrations of circulating megakaryocyte DNA (average 250 GE/ml, $p < 0.05$). These findings are consistent with megakaryocyte cfDNA levels being reflective of megakaryocyte function and turnover (Fig. 5A, B). Notably, erythroblast cfDNA was not reflective of megakaryocyte-platelet pathologies but was significantly elevated in the plasma of patients with thalassemia major, a disease involving increased, albeit

ineffective, production of erythrocytes within the bone marrow (average of 1658 erythroblast GE/ml in thalassemia patients compared with 7 GE/ml in healthy individuals, $p < 0.05$) (Fig. 5C). These findings confirm the specificity of our targeted markers and support their potential utility in highly specific identification of altered turnover of the respective cell types of origin.

## Discussion

In this work, we demonstrated that megakaryocytes are major contributors to plasma cfDNA in healthy individuals, giving rise to ~26% of total cfDNA. In contrast, erythroblasts give rise to a small but distinct proportion (~0.5–4%) of cfDNA in healthy individuals. These findings contrast with previous reports that identified a major contribution of erythroblasts to cfDNA of healthy individuals. One of these studies, from our own group, relied on a tissue methylome atlas (based on Illumina methylation arrays) which contained the methylome of common megakaryocyte-erythrocyte progenitor cells but not the megakaryocyte methylome[12]. Another study has demonstrated that genomic loci uniquely unmethylated in erythroblasts are also partially unmethylated in plasma, concluding that erythroid cells are a major source of cfDNA in healthy people[13]. A close examination of the methylation status of these marker loci revealed that they are in fact unmethylated not only in erythroblasts but also in megakaryocytes (Supplementary Fig. S12). Thus, these markers are not sufficient to distinguish between erythroid and megakaryocyte contributions to plasma, and the presence of unmethylated DNA from these loci in healthy plasma is in fact consistent with a significant presence of megakaryocyte DNA in circulation, in line with our current findings and as suggested by analysis of histone modifications in circulating chromatin[11].

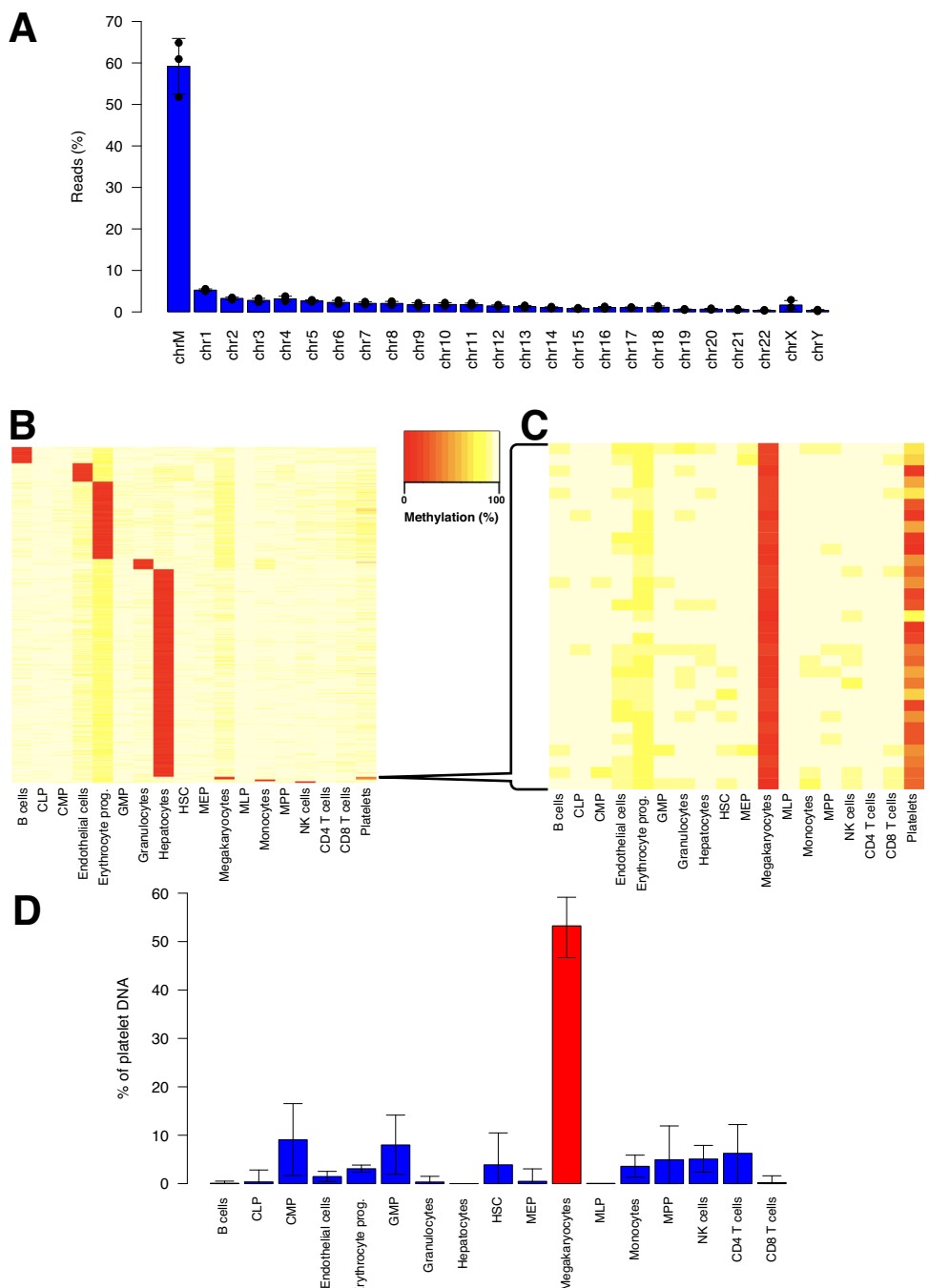

**Fig. 3 | Genome-wide analysis of platelet DNA supports megakaryocyte origin.** **A** Platelet DNA is derived from all human chromosomes; however, the majority is mitochondrial DNA. The mean of three platelet samples is plotted with error bars representing standard deviations. **B**, **C** Regions uniquely unmethylated in cell types of hematopoietic origin, hepatocytes and endothelial cells were identified as described (Methods). Megakaryocyte-unique unmethylated regions are unmethylated in platelets as well. HSC Hematopoietic stem cell, MPP Multipotent progenitor, CMP Common myeloid progenitor, CLP Common lymphoid progenitor, MLP Multi-lymphoid progenitor, GMP Granulocyte macrophage progenitor, MEP Megakaryocyte erythroid progenitor. **D** Deconvolution of platelet DNA methylation demonstrated megakaryocyte DNA as the main component of platelet DNA. Error bars represent 90% confidence intervals calculated by bootstrapping over 10000 iterations of pooled platelet samples. The bar height represents the average across the bootstrapped samples. Source data are provided as a Source Data file.

Megakaryocytes and erythroblasts have distinct mechanisms of cfDNA release. Megakaryocytes are thought to undergo apoptosis during or immediately subsequent to thrombopoiesis[25]. Our analysis indicates that while a small proportion of the megakaryocyte genome (~0.1%) is trapped within platelets, a fraction is fragmented to nucleosome-size pieces and released to circulation as cfDNA. As for erythroblasts, the process of nuclear extrusion in the final differentiation step of erythrocytes involves efficient removal by local macrophages[26]. Indeed, while 214 billion erythroblast nuclei are extruded every day[27], only a small amount of erythroblast DNA is present in plasma. More experiments are needed to define the exact origin of erythroblast cfDNA.

Additionally, we demonstrated that megakaryocyte- and erythroblast-specific cfDNA concentration is reflective of megakaryocyte and red blood cell precursor activity; thus, it can serve as a biomarker for diseases involving aberrant activity of these

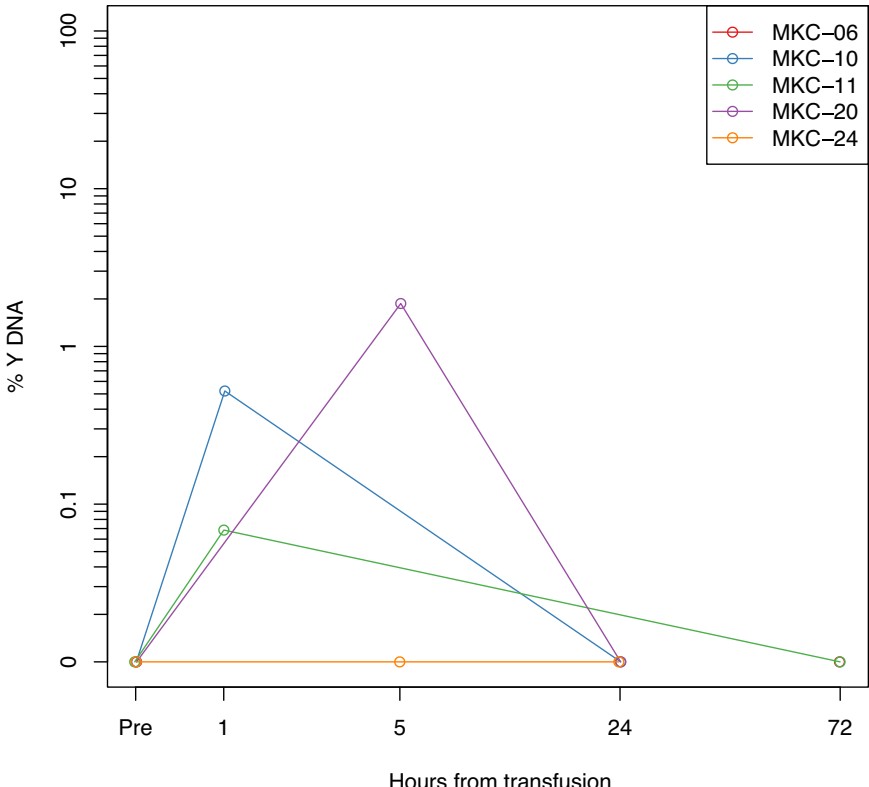

**Fig. 4 | Sex-mismatched platelet transfusions suggest that platelets are not the source of megakaryocyte DNA.** The ratio of concentration of DNA from the SRY gene, located on the Y-chromosome, to DNA from the Beta-actin gene, located on chromosome 7, was evaluated in the plasma of female recipients of platelets from male donors, by massive parallel sequencing. At 24–72 h after transfusion there was no remaining male DNA detectable. Each line represents a different individual. Source data are provided as a Source Data file.

**Table 1 | Description and results of clinical samples analyzed for megakaryocyte and erythroblast cfDNA concentrations**

| Group | Count | Age (avg.) | Age (SD) | Female (n) | Platelet count (avg. n/ul) | Megakaryocyte cfDNA (avg. GE/ml) | Erythroblast cfDNA (avg. GE/ml) |
|---|---|---|---|---|---|---|---|
| **ITP** | 5 | 42 | 25 | 3 | 14400 | 1608 | 21 |
| **ET** | 5 | 67 | 15 | 3 | 1064600 | 4242 | 112 |
| **Hypoplastic BM** | 5 | 60 | 21 | 2 | 39600 | 250 | 2126 |
| **Healthy** | 77 | 36 | 15 | 40 | 302545 | 510 | 7 |
| **Thalassemia** | 3 | 48 | 7 | 2 | 619667 | 690 | 1658 |

blood lineages, not necessarily reflected in peripheral platelet and RBC counts.

Thrombocytopenia in cancer patients can be induced by chemotherapy, bone marrow involvement or peripheral destruction of platelets as a result of immune processes.

Prolonged thrombocytopenia in cancer patients during therapy can be a diagnostic challenge. Bone marrow biopsy is used for differential diagnosis, but the biopsy may be only partially representative. Determining the cause of the thrombocytopenia is crucial, as treatment for such thrombocytopenic patients will differ according to the pathogenesis of the thrombocytopenia.

Combined with peripheral blood smear and bone marrow biopsy, cfDNA measurements can potentially distinguish between these clinical scenarios and guide treatment decisions.

ITP is a syndrome characterized by antibody-mediated platelet destruction and variably reduced platelet production[28]. Bone marrow biopsies of ITP patients show normal or increased numbers of megakaryocytes. There is currently no diagnostic tool for ITP. We demonstrated that megakaryocyte cfDNA is elevated in ITP. More

experiments are needed in ITP patients and healthy controls to determine whether megakaryocyte methylation biomarkers can serve as a diagnostic tool for ITP.

This study also provides evidence that platelets contain megakaryocyte-derived nuclear DNA. An important implication of this situation is that it provides a non-invasive window into the genome and epigenome of the megakaryocyte, a cell type particularly difficult to interrogate due to its relative scarceness within the bone marrow. The presence of somatic mutations in megakaryocytes, as well as epigenomic aberrations, can therefore be interrogated by analysis of platelet samples. Additionally, gene expression of the megakaryocyte, as reflected by transcription related epigenetic modifications, may be indirectly evaluated via the platelet epigenome. Interestingly, we found copy number alterations in the DNA of platelets, with some chromosomes consistently over- or under-represented. We speculate that this situation reflects the 3D organization of the megakaryocyte genome, affecting the likelihood that certain genomic regions will be trapped in platelets. Finally, our results suggest that megakaryocyte cfDNA is derived directly from megakaryocytes and

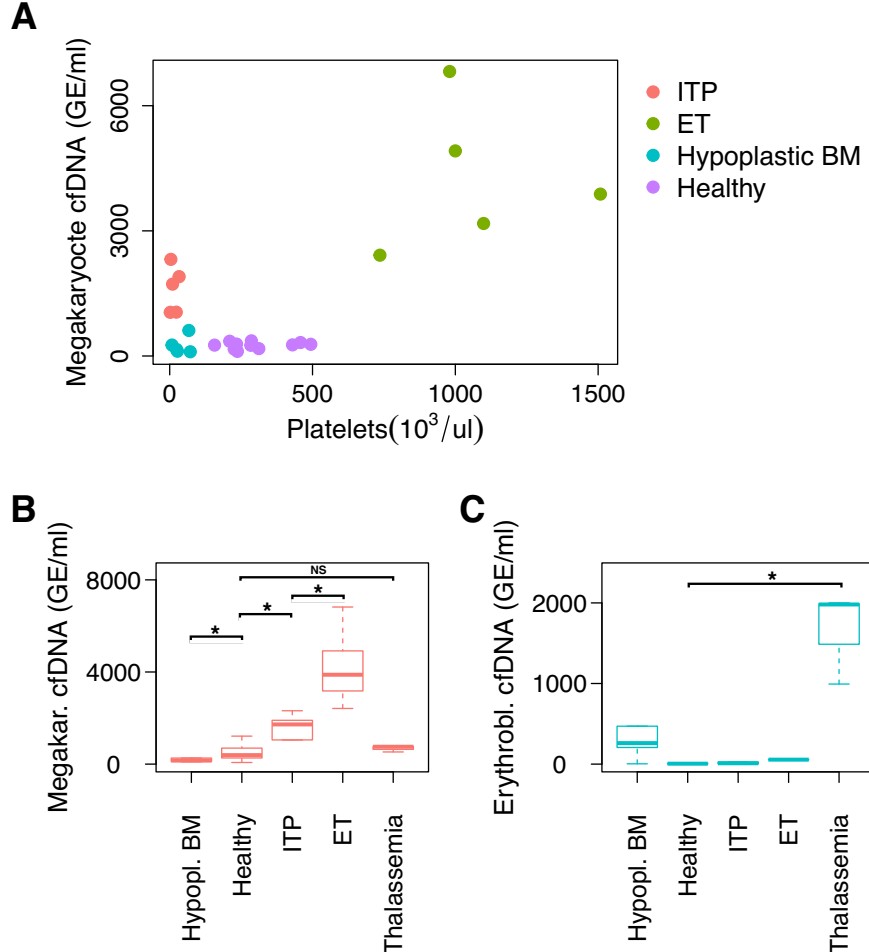

**Fig. 5 | Targeted analysis of megakaryocyte (MK) and erythroblast methylation markers in plasma samples. A** Genome equivalents (GE) per milliliter plasma of megakaryocyte DNA in samples from healthy donors and patients with Idiopathic Thrombocytopenic Purpura (ITP), ET Essential Thrombocythemia (ET) and hypoplastic bone marrow (after chemotherapy). **B** MK DNA is present in significantly different concentrations between healthy ($n = 61$), ITP ($n = 5$), ET ($n = 5$) and hypoplastic bone marrow ($n = 5$) ($p < 0.05$). MK DNA is not significantly different in thalassemia ($n = 3$) as compared to healthy individuals. All samples are biologically

independent samples. The center, bounds of box and whiskers represent the median, upper and lower quartiles, and 1.5x the interquartile range, respectively. **C** Erythroblast cfDNA is significantly elevated in thalassemia ($n = 3$) compared to healthy individuals ($n = 70$), as well as compared to ET ($n = 5$) ($p < 0.05$). Data is plotted for hypoplastic bone marrow ($n = 5$) and ITP ($n = 5$) as well. All samples are biologically independent samples. The center, bounds of box and whiskers represent the median, upper and lower quartiles, and 1.5x the interquartile range, respectively. *$p < 0.05$. Source data are provided as a Source Data file.

not from platelets. This conclusion is based on the lack of Y chromosome fragments in the plasma of female recipients of male platelets; on the fact that copy number alterations seen in platelets are not present in cfDNA; and on the finding that platelets capture only 0.1% of the DNA of the parent cell. Further studies are required to validate this conclusion and understand the underlying mechanisms. It is important to note that this study was performed in individuals without evidence of platelet hyperactivation. It is possible that during platelet activation, DNA is indeed released directly from platelets. If this is the case, platelet-derived DNA may be an important analyte for the study of hypercoagulable states. This should be a focus of future research.

We have not addressed whether platelet DNA is concentrated in a subset of platelets or if it is distributed randomly throughout the platelet population. Additionally, the fate and function of the megakaryocyte genomic fragments trapped within platelets is not clear. It is well established that platelets contain RNA which undergoes translation to protein, yet this RNA is generally understood to have been transcribed in the megakaryocyte prior to platelet formation or from mitochondrial DNA within the platelet[29]. Given our findings that platelets contain nuclear DNA, the question as to whether

transcription of nuclear DNA occurs within the platelet should be revisited.

## Methods
### Subject enrollment
The study protocol was approved by the Committee on Research Involving Human Subjects of the Hebrew University-Hadassah Medical School, Jerusalem, Israel, with procedures performed in accordance with the Declaration of Helsinki. Blood samples were obtained from donors who have provided written informed consent. Subject characteristics are presented in Supplementary Table S3. Patients were recruited with acute ITP, with less than 35000 platelets/ul ($n = 5$), with Essential Thrombocytopenia ($n = 5$, >500000 platelets/ul); 5 patients with Acute Myeloid Leukemia (AML), 14 days after the start of induction chemotherapy with cytarabine and daunorubicin (7 + 3 regimen) were recruited as patients with hypoplastic bone marrow. Additionally, 3 patients were recruited with thalassemia major. 77 healthy controls were recruited. 11 female individuals who received platelet transfusions from male donors were recruited. Healthy controls were excluded if platelet count was less than 150000 platelets/ul or greater than 500000 platelets/ul.

## Platelet isolation

Platelets were isolated from donors via single donor plateletpheresis for the purpose of platelet transfusion at the Hadassah Medical Center Blood Bank. Platelets which had not been transfused within one week were obtained for research purposes. In order to subsequently purify platelets from surrounding plasma, the platelets were diluted in PBS noCa$^{2+}$ & Mg$^{2+}$ or tyrode's buffer (1/3 SDP + 2/3 PBS). Platelet activation was inhibited with citric acid (5ul of 1 M citric acid per 1 ml or 10% citrate from collection tubes). Subsequently, centrifugation was performed at 750 g (3000 rpm) for 8 min at room temperature. A platelet pellet was obtained which was diluted in PBS with citric acid (5 ul citric acid/ml PBS). Optionally, Dnase I was added at a concentration of 100μg/ml and samples were incubated for 20 min in a heat bath at 37 degrees Celsius. Samples were then centrifuged again at 750 g for 8 min and then pellets were suspended in 1.4 ml PBS (no citrate).

## Blood sample collection and processing

Blood samples were collected by routine venipuncture in 10 ml EDTA Vacutainer tubes or Streck blood collection tubes and stored at room temperature for up to 4 hr or 5 days, respectively. Tubes were centrifuged at 1500× g for 10 min at 4 °C (EDTA tubes) or at room temperature (Streck tubes). The supernatant was transferred to a fresh 15 ml conical tube without disturbing the cellular layer and centrifuged again for 10 min at 3000× g. The supernatant was collected and stored at –80 °C. cfDNA was extracted from 2 to 4 ml of plasma using the QIAsymphony liquid handling robot (Qiagen). cfDNA concentration was determined using Qubit double-strand molecular probes kit (Invitrogen) according to the manufacturer's instructions.

DNA derived from all samples was treated with bisulfite using EZ DNA Methylation-Gold (Zymo Research), according to the manufacturer's instructions, and eluted in 24 μl elution buffer.

For preparing PRP, whole blood (anticoagulated with 3.2 g% of sodium citrate, ratio 1:10) was centrifuged at 110 g for 10 min at room temperature. 0.5 ml aliquot was collected for evaluation. The remaining PRP was diluted in PBS noCa2+ & Mg2+ (1/3 PRP + 2/3 PBS) and centrifuged at 960 g for 10 min, to collect platelets. The platelet pellet was resuspended in 100ul PBS for evaluation.

## Whole-genome bisulfite sequencing of reference cell types

Previously published single-cell genome-wide methylation profiles of megakaryocytes and genome-wide profiles of megakaryocyte-erythrocyte progenitors[30] of three individuals were obtained from Blueprint Epigenome. Single-cell methylation profiles were grouped by individual for further analysis.

Erythroblasts were isolated as follows: Bone marrow was diluted with 3 equivalent volumes of PBS, filtered through a 100um cell strainer and laid over Lymphoprep density gradient medium (Stemcell Technologies). After centrifugation, interphase bone marrow mononuclear cells were transferred to PBS, washed and red blood cells (RBC) were lysed using standard RBC lysis buffer.

CD45-negative, CD235a and CD71-positive erythrocyte precursors were FACS-sorted on a BD FACSAriaTM III flow cytometer.

Granulocytes, monocytes, B cells, NK cells, CD3 + T cells, CD4 + T cells, CD8 + T cells, hepatocytes and endothelial cells were isolated as previously described[31].

Up to 75 ng of sheared gDNA was subjected to bisulfite conversion using the EZ-96 DNA Methylation Kit (Zymo Research; Irvine, CA), with liquid handling on a Hamilton MicroLab STAR (Hamilton; Reno, NV). Dual indexed sequencing libraries were prepared using Accel-NGS Methyl-Seq DNA library preparation kits (Swift BioSciences; Ann Arbor, MI) and custom liquid handling scripts executed on the Hamilton MicroLab STAR. Libraries were quantified using KAPA Library Quantification Kits for Illumina Platforms (Kapa Biosystems; Wilmington, MA). Four uniquely dual indexed libraries, along with 10% PhiX v3 library (Illumina; San Diego, CA), were pooled and clustered on a

Illumina NovaSeq 6000 S2 flow cell followed by 150-bp paired-end sequencing.

## Selection of megakaryocyte and erythroblast methylation markers

CpGs were identified as being cell type specific if they were unmethylated (<20% methylation) in all samples of the cell type of interest and methylated in all other samples (>80% methylation) or if they were methylated in all samples of the cell type of interest and unmethylated in other samples.

For selection of markers for analysis by targeted bisulfite sequencing, CpGs with neighboring CpGs which were also cell type specific and located in a CpG dense genomic region (>=5 CpGs within 120 bp) were selected for further analysis. DNA methylation of candidate regions were then compared to DNA methylation at these regions in genome-wide methylation profiles of multiple tissues published as part of Roadmap Epigenomics, using the same criteria as described above.

Genome coordinates of cell-type specific methylation markers and primer sequences are included in Supplementary Table S1.

## PCR

To efficiently amplify and sequence multiple targets from bisulfite-treated cfDNA, we used a two-step multiplexed PCR protocol[32]. Primer sequences are included in Supplementary Table S1. In the first step, up to 10 primer pairs were used in one PCR reaction to amplify regions of interest from bisulfite-treated DNA, independent of methylation status. Primers were 18–30 base pairs (bp) with primer melting temperature ranging from 58 °C to 62 °C. To maximize amplification efficiency and minimize primer interference, the primers were designed with additional 25 bp adapters comprising Illumina TruSeq Universal Adapters without index tags. All primers were mixed in the same reaction tube. For each sample, the PCR was prepared using the QIAGEN Multiplex PCR Kit according to manufacturer's instructions with 7 μl of bisulfite-treated cfDNA. Reaction conditions for the first round of PCR were: 95 °C for 15 min, followed by 30 cycles of 95 °C for 30 s, 57 °C for 3 min and 72 °C for 1.5 min, followed by 10 min at 68 °C.

In the second PCR step, the products of the first PCR were treated with Exonuclease I (ThermoScientific) for primer removal according to the manufacturer's instructions. Cleaned PCR products were amplified using one unique TruSeq Universal Adapter primer pair per sample to add a unique index barcode to enable sample pooling for multiplex Illumina sequencing. The PCR was prepared using 2× PCRBIO HS Taq Mix Red Kit (PCR Biosystems) according to manufacturer's instructions. Reaction conditions for the second round of PCR were: 95 °C for 2 min, followed by 15 cycles of 95 °C for 30 s, 59 °C for 1.5 min, 72 °C for 30 s, followed by 10 min at 72 °C. The PCR products were then pooled, run on 3% agarose gels with ethidium bromide staining, and extracted by Zymo GEL Recovery kit.

## NGS and analysis of PCR products

Pooled PCR products were subjected to multiplex NGS using the *NextSeq* 500/550 v2 Reagent Kit (Illumina). Sequenced reads were separated by barcode, and aligned to the target sequence with Bismark, using a computational pipeline available (https://github.com/Joshmoss11/btseq). CpGs were considered methylated if 'CG' was read and unmethylated if 'TG' was read. Proper bisulfite conversion was assessed by analyzing methylation of non-CpG cytosines. We then determined the fraction of molecules in which all CpG sites were unmethylated. The fraction obtained was multiplied by the concentration of cfDNA measured in each sample, to obtain the concentration of tissue-specific cfDNA from each donor. Given that the mass of a haploid human genome is 3.3 pg, the concentration of cfDNA could be converted from units of ng/ml to haploid GE/ml by multiplying by a factor of 303.

Sequenced molecules were considered to be unmethylated (for specifically unmethylated markers) if all CpGs on the sequenced DNA fragment were unmethylated for erythroblast markers and if all CpGs, with the exception of up to one methylated CpG, were unmethylated for megakaryocyte markers. The fraction of DNA derived from the cell type of interest for each marker (for specifically unmethylated markers) was calculated as the fraction of completely unmethylated molecules among all sequenced molecules from the marker region.

### Whole-genome bisulfite sequencing of platelet DNA

DNA was extracted from three platelet samples using the QIAsymphony liquid handling robot (Qiagen). Subsequently, whole-genome bisulfite sequencing libraries were generated with Swift AccelNGS Methyl-Seq DNA Library preparation protocol (Swift Biosciences, Ann Arbor, MI). Paired-end sequencing was performed on the Illumina Nextseq 550 System of 300 bp per read at an average depth of 2.54x.

### Whole-genome bisulfite sequencing computational processing and analysis

Paired-end FASTQ files were mapped to the human (hg19) genome using bwa-meth (V 0.2.0), with default parameters[33], then converted to BAM files using SAMtools (V 1.9)[34]. Duplicated reads were marked by Sambamba (V 0.6.5), with parameters "-l 1 -t 16 --sort-buffer-size 16000 --overflow-list-size 10000000"[35]. Reads with low mapping quality, duplicated, or not mapped in a proper pair were excluded using SAMtools view with parameters -F 1796 -q 10. Reads were stripped from non-CpG nucleotides and converted to BETA files using wgbstools (V 0.1.0) (https://github.com/nloyfer/wgbs_tools)[31].

For comparison of platelet DNA methylation to DNA methylation of other cell types, bone marrow-residing immune cell progenitor cells were obtained from Blueprint Epigenome[36]. Genome-wide methylation profiles of hepatocytes, endothelial cells and immune cells, which have been found to release DNA to plasma under normal conditions[12], were obtained[31]. Regions with methylation unique to each cell type were identified by segmenting the genome into multi-sample homogenous blocks as previously described[31], and identifying regions unmethylated (<20%) or methylated (>80%) in a specific cell type. Deconvolution of the platelet DNA methylation profiles at the resultant regions was performed by NNLS as previously described[12].

### Statistics

To determine the significance of differences between groups we used a non-parametric two-tailed Mann-Whitney test. P-values were considered significant at <0.05. Samples that were detected as outliers were excluded. All statistical analyses were performed with R (version 4.1)[37]. In bar plots, standard deviation is represented by error bars. For box plots, the center line, box limits and whiskers represent the median, upper and lower quartiles, and 1.5x the interquartile range, respectively.

### Reporting summary

Further information on research design is available in the Nature Portfolio Reporting Summary linked to this article.

## Data availability

Source data are provided with this paper. The whole-genome bisulfite sequencing (WGBS) data of platelet DNA generated in this study have been deposited in the GEO database under accession code GSE206818. The raw sequencing data are protected and are not available due to data privacy laws. The targeted bisulfite sequencing data generated in this study are provided in the Supplementary Information/Source Data file. The bone-marrow residing progenitor cells WGBS data used in this study are available in the IHEC data portal [https://epigenomesportal.ca/ihec/]. WGBS data for erythroblasts, leukocytes, hepatocytes, endothelial cells used in this study are available in the GEO database

under accession code GSE186458. Source data are provided with this paper.

## Code availability

Whole genome bisulfite sequencing analysis was performed using wgbstools (V 0.1.0) (https://github.com/nloyfer/wgbs_tools). Targeted bisulifte sequencing analysis was performed using btseq (https://github.com/Joshmoss11/btseq).

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

## Acknowledgements
We acknowledge the donors of the biological samples presented in this study for their benevolent contribution to science. We thank Dr. Abed Nasereddin and Dr. Idit Shiff of the Core Research Facility of the Faculty of Medicine of the Hebrew University of Jerusalem for performing the targeted bisulfite sequencing experiments and WGBS of platelets presented in the article, and the blood bank of the Hadassah Medical Center for aid in obtaining platelet samples. We thank Prof. Neta Goldschmidt for recruiting patients to the study. This study was supported by grants from The Ernest and Bonnie Beutler Research Program of Excellence in Genomic Medicine (FREEDOME) (to Y.D.), The Helmsley Charitable Trust (to Y.D.), Alzheimer's Drug Discover Foundation (to Y.D.), a Merck Grant for Multiple Sclerosis Innovation (GMSI) (to Y.D.), The Israel Science Foundation (to Y.D.), the Waldholtz/Pakula family (to Y.D.), and the Robert M. and Marilyn Sternberg Family Charitable Foundation (to Y.D.). Y.D. holds the Walter and Greta Stiel Chair and Research grant in Heart studies.

## Author contributions
Conceptualization was performed by J.M., Y.D., R.S., A.A., and R.B. Methodology was performed by J.M., R.B., Y.D., G.C., B.G., R.S., A.K., E.S., and Y.K. Investigation was performed by J.M., R.B., A.K., A.A., E.S., and O.G.. Visualization was performed by J.M. and A.K. Funding acquisition was performed by Y.D. Project administration was performed by Y.D., R.S., A.A., and J.M. Supervision was performed by Y.D., R.S., and A.A. Writing – original draft was performed by J.M. Writing – review & editing was performed by J.M., Y.D., B.G., A.A., Y.K., R.B., G.C., R.S.

## Competing interests
J.M., R.S., B.G., and Y.D. are inventors of patents describing methylation markers and their use for cfDNA analysis. All remaining authors have declared no conflicts of interest.
