## [Peer Review File · Nature Communications]

Megakaryocyte- and erythroblast-specific cell-free DNA patterns in plasma and platelets reflect thrombopoiesis and erythropoiesis levelsREVIEWER COMMENTS

Reviewer #1 (Remarks to the Author):

In this manuscript, Moss et al report a series of compelling experiments to characterize megakaryocyte and erythroblast DNA in plasma and platelets.

The authors used innovative methods, based on analyses of cell-type specific methylation marks, to quantify the contributions of megakaryocyte and erythroblast DNA in plasma of healthy individuals, and in patients with pathologies involving increased platelet production.

They report the surprising and novel finding that platelets contain genomic DNA, and they show this DNA is derived from megakaryocytes. In addition to this observation of new biology, the authors demonstrate significant potential for translational impact of their findings and techniques, with the demonstration that megakaryocyte and erythroblast specific DNA can serve as biomarkers for diseases that are associated with altered thrombopoiesis and erythropoiesis.

The manuscript is very well written, the analysis easy to follow, and the data are presented in a clear manner. I am a fan of this paper. I am very happy to recommend it for publication without modification.

My only suggestion would be to analyze the genomic origin of the DNA in platelets in a bit more detail. The authors show that all chromosomes are represented but are all parts of all chromosomes found in platelets, or is there an enrichment of specific parts of the genome?

Reviewer #2 (Remarks to the Author):

This is a very interesting paper that continues on work done by the Dor lab to use normal tissue specific DNA methylation patterns to ask interesting biological and clinical questions. This paper deals with the tissue origin of cell free DNA in the blood in terms of the principle cells of origin and helps to correct or clarify previous publications (including their own) that had identified erythroid cells as one of the principle sources. As Dor clearly demonstrates Lam et al had used a methylation marker that did not differentiate between megakaryocyte and erythroid cells. This paper identifies a definitive signature of megakaryocytes and clearly demonstrates the major contribution of this cell type to the totality of cfDNA. This in itself is an interesting observation but one not necessarily compelling enough to warrant

publication in Nat Comm. However, they do demonstrate what could be multiple examples of clinical utility, but again these are fairly specific and relatively low frequency diseases, but which are none the less important. The most novel observation is the presence of megakaryocyte DNA in platelets. They characterize this DNA quite extensively and provide insight into whether it may be the origin of the megakaryocyte cfDNA. The sex difference platelet transfusion experiment is very elegant and while mostly convincing I would like to see additional support for this experiment. They are relying on the persistence of transfused platelets for more than 2 days and the lack of the Y chromosome signal in plasma as proof of the lack of contribution to cfDNA. Demonstration of the persistence of the Y chromosome DNA in platelets at the later time points and how much of the population of platelets is from transfused platelets, would reinforce the conclusion. Otherwise it is not clear to me that transfused platelets are abundant enough to serve as a detectable source of cfDNA. It is beholding on them to demonstrate this. If this is done I would definitely recommend this paper for publication.

Reviewer #3 (Remarks to the Author):

In this manuscript, Moss and colleagues suggest that anuclear platelets are major contributors to cfDNA. Further they say that their data support the finding that the genomic DNA fragments found in platelet originate in megakaryocytes. While the topic is intriguing and of significant interest to myself and the general field, this manuscript unfortunately has fatal flaws that preclude me from recommending it for publication. The first is that previously published single-cell genome-wide methylation profiles of megakaryocytes and genome-wide profiles of megakaryocyte-erythrocyte progenitors were used from Farlik et al (Cell Stem Cell 2016). This was combined with newly generated data from other cell types. I do not think that this is appropriate, as these data were generated in 2016. First, the technology available in 2016 versus now has changed significantly. Further, there are multitude experimental conditions that vary even in within replicates in the same lab. It is impossible to think conditions could be similar between labs, years apart.

The other main concern is that in the 2016 Farlik paper, they sorted megakaryocytes by ploidy and found that a small number of region sets were differentially methylated, and these regions underwent consistent and progressive changes across the different ploidy stages of megakaryocyte maturation. Therefore, this suggests that the methylation profile of megakaryocytes is indeed not consistent, but varies by ploidy. This then undermines the premise of the manuscript. If the megakaryocyte methylome is not consistent, how can it be faithfully used to fingerprint cfDNA in platelets?

Detailed responses to Reviewer comments

We thank the reviewers for their overall positive comments. We have revised the manuscript accordingly, including experimental and text revisions. The highlights of the revisions are:

- The unexpected observation that the megakaryocyte genome is unevenly represented in platelet DNA, with some chromosomal regions more likely, and others less likely, to be present in platelets. We report this observation and speculate on the potential underlying mechanism.
- Additional experiments with male platelets transfused to female recipients, which lend further support to the idea that MK, rather than platelets, are the main origin of abundant MK cfDNA.
- Additional comparative analysis of previously published MK methylomes and current methylomes of platelets and other cell types, supporting the finding that platelet DNA originates in MK.

Reviewer #1 (Remarks to the Author):

In this manuscript, Moss et al report a series of compelling experiments to characterize megakaryocyte and erythroblast DNA in plasma and platelets.

The authors used innovative methods, based on analyses of cell-type specific methylation marks, to quantify the contributions of megakaryocyte and erythroblast DNA in plasma of healthy individuals, and in patients with pathologies involving increased platelet production.

They report the surprising and novel finding that platelets contain genomic DNA, and they show this DNA is derived from megakaryocytes. In addition to this observation of new biology, the authors demonstrate significant potential for translational impact of their findings and techniques, with the demonstration that megakaryocyte and erythroblast specific DNA can serve as biomarkers for diseases that are associated with altered thrombopoiesis and erythropoiesis.

The manuscript is very well written, the analysis easy to follow, and the data are presented in a clear manner. I am a fan of this paper. I am very happy to recommend it for publication without modification.

My only suggestion would be to analyze the genomic origin of the DNA in platelets in a bit more detail. The authors show that all chromosomes are represented but are all parts of all chromosomes found in platelets, or is there an enrichment of specific parts of the genome?

Response: We thank the reviewer for general support of the manuscript and for the helpful suggestion. After analyzing the relative representation of different genomic regions in DNA extracted from platelets, we found that the relative frequency of chromosomes is in fact not random. Some chromosomes (ex. chr17) are under-represented, while other chromosomes (ex. chr4) are enriched. This is an intriguing finding which we hypothesize may be related to the probability of certain nuclear regions in a megakaryocyte to enter a platelet during thrombopoiesis. As this finding is not directly related to the central focus of the manuscript, we propose that further investigation as to the related mechanisms should be pursued in future studies. We have now addressed this in the manuscript (line 172 and new Supplementary figure S7). However, we did use this finding to support the argument that MK cfDNA likely originates directly from MK, rather than from platelets.

Reviewer #2 (Remarks to the Author):

This is a very interesting paper that continues on work done by the Dor lab to use normal tissue specific DNA methylation patterns to ask interesting biological and clinical questions. This paper deals with the tissue origin of cell free DNA in the blood in terms of the principle cells of origin and helps to correct or clarify previous publications (including their own) that had identified erythroid cells as one of the principle sources. As Dor clearly demonstrates Lam et al had used a methylation marker that did not differentiate between megakaryocyte and erythroid cells. This paper identifies a definitive signature of megakaryocytes and clearly demonstrates the major contribution of this cell type to the totality of cfDNA. This in itself is an interesting observation but one not necessarily compelling enough to warrant publication in Nat Comm. However, they do demonstrate what could be multiple examples of clinical utility, but again these are fairly specific and relatively low frequency diseases, but which are none the less important. The most novel observation is the presence of megakaryocyte DNA in platelets. They characterize this DNA quite extensively and provide insight into whether it may be the origin of the megakaryocyte cfDNA. The sex difference platelet transfusion experiment is very elegant and while mostly convincing I would like to see additional support for this experiment. They are relying on the persistence of transfused platelets for more than 2 days and the lack of the Y chromosome signal in plasma as proof of the lack of contribution to cfDNA. Demonstration of the persistence of the Y chromosome DNA in platelets at the later time points and how much of the population of platelets is from transfused platelets, would reinforce the conclusion. Otherwise it is not clear to me that transfused platelets are abundant enough to serve as a detectable

source of cfDNA. It is beholding on them to demonstrate this. If this is done I would definitely recommend this paper for publication.

Response: We thank the reviewer for their interest in our discoveries. We share their appreciation of the novel discovery of DNA in platelets, and accept their suggestion to reinforce the claim that circulating megakaryocyte DNA in plasma is derived from megakaryocytes and not from platelets.

We have exerted great effort to address this issue and this has been a main focus of our work recently. In the revised manuscript we provide three independent lines of evidence to support the idea MK cfDNA originates in MK rather than platelets:

1. We have collected blood from women 24 hours after platelet transfusion from men, and assessed the presence of Y chromosome-derived DNA in both plasma and platelets. Importantly, while in some cases (2/4) we identified Y chromosome DNA in platelets, in no cases did we find Y chromosome DNA in circulating cell free DNA. This supports the claim that circulating MK DNA is not derived from platelets. The data are shown in Figure 4 and Supplementary Figures S10, S11.

2. We discovered chromosomal imbalance in the DNA present within platelets (in response to reviewer #1, new Supplementary Figure S7), which is not seen in megakaryocyte genomic DNA. If platelets were the main source of MK cfDNA (amounting to 26% of cfDNA), the imbalance should have been observed in genomic analysis of cfDNA. However, cfDNA contains a balanced representation of genomic DNA. This further suggests that cfDNA originates in MK rather than platelets.

3. We found that platelets generated by one megakaryocyte contain only a small fraction of the genome of that cell (Supplementary Table S2). This makes it more likely that MK cfDNA originates in dying MK (containing 99.9% of the MK genome) rather than in platelets (containing 0.1% of the MK genome).

We are aware that the claim that MK cfDNA is derived directly from megakaryocytes is based on a small number of samples, and should be validated with further data. We have been cautious to not use overly assertive language (line 197) and have stressed these reservations in the discussion (line 301).

Reviewer #3 (Remarks to the Author):

In this manuscript, Moss and colleagues suggest that anuclear platelets are major contributors to cfDNA.

Response: In fact, we concluded (and this conclusion is supported by further experiments in the revised manuscript) that platelets are NOT a major contributor to cfDNA. Our analysis suggests that megakaryocytes directly release DNA to plasma, which accounts for the 26% of total cfDNA.

Further they say that their data support the finding that the genomic DNA fragments found in platelet originate in megakaryocytes. While the topic is intriguing and of significant interest to myself and the general field, this manuscript unfortunately has fatal flaws that preclude me from recommending it for publication. The first is that previously published single-cell genome-wide methylation profiles of megakaryocytes and genome-wide profiles of megakaryocyte-erythrocyte progenitors were used from Farlik et al (Cell Stem Cell 2016). This was combined with newly generated data from other cell types. I do not think that this is appropriate, as these data were generated in 2016. First, the technology available in 2016 versus now has changed significantly. Further, there are multitude experimental conditions that vary even within replicates in the same lab. It is impossible to think conditions could be similar between labs, years apart.

Response: We thank the reviewer for the detailed remarks and careful attention to detail.

The reviewer is concerned that comparison of megakaryocyte methylomes (generated in 2016) with more recently generated data from other cell types is not legitimate due to differences in the technology of methylome analysis since 2016.

We respectfully disagree. While technology has evolved, the 2016 study used the same Illumina NGS technology that we are using today. More importantly, an unmethylated region in the same cell type should have been detected equally robustly in 2016 and today. Nonetheless, in order to assess compatibility of the 2016 and 2023 data, we have reanalyzed cell type specific markers used in the 2016 study with datasets originating in the 2016 study as well as datasets derived from the same cell types, isolated in our laboratory. We found that samples of cells for which markers were identified originating in the 2016 study are highly correlated with samples of the same cell types recently generated. This is now shown in new Supplementary Figure S8.

The other main concern is that in the 2016 Farlik paper, they sorted megakaryocytes by ploidy and found that a small number of region sets were differentially methylated, and these regions underwent consistent and progressive changes across the different ploidy stages of megakaryocyte maturation. Therefore, this suggests

that the methylation profile of megakaryocytes is indeed not consistent, but varies by ploidy. This then undermines the premise of the manuscript. If the megakaryocyte methylome is not consistent, how can it be faithfully used to fingerprint cfDNA in platelets?

Response: The reviewer points out that megakaryocyte methylation changes with ploidy and therefore the methylation 'fingerprint' may not actually represent all megakaryocytes. We have now reanalyzed the regions found to have megakaryocyte unique methylation patterns, this time separated by ploidy. While some of the markers do appear to change with ploidy, overall the markers clearly distinguish MK samples of any ploidy from other cell types, and show the similarity of MK DNA to platelet DNA. This is shown in new Supplementary Figure S9 and mentioned in the manuscript (line 179).

REVIEWER COMMENTS

Reviewer #2 (Remarks to the Author):

The authors have addressed the issues that were outstanding from my review, improving the work significantly. The manuscript is now acceptable for publication.

Reviewer #3 (Remarks to the Author):

The authors have done an excellent job responding to the concerns that were raised, and I agree with their comments.

My only lasting concern is the assertion that the cfDNA is released from the MK and not from the platelet. This is counterintuitive because when platelets activate, they release their contents. So, how would the cfDNA be retained, and not released into circulation, once platelets are activated? Is it packaged into a novel compartment of a platelet that is not a granule? If so, this should be shown directly.

Reviewer #3 remarks:

The authors have done an excellent job responding to the concerns that were raised, and I agree with their comments.

My only lasting concern is the assertion that the cfDNA is released from the MK and not from the platelet. This is counterintuitive because when platelets activate, they release their contents. So, how would the cfDNA be retained, and not released into circulation, once platelets are activated? Is it packaged into a novel compartment of a platelet that is not a granule? If so, this should be shown directly.

Author response:

We agree with the reviewer that activated platelets, may in fact cause release of platelet DNA into circulation, although release of granules from platelets is a highly regulated process, and therefore it is not clear that DNA would indeed be released during the process. Nevertheless, this is a very interesting suggestion and implies that platelet cfDNA may be an important analyte for studying platelet activation. However, under normal homeostasis, platelets are not regularly activated and therefore we would not expect to detect DNA derived from activated platelets in plasma. In this study, we evaluated situations without clear evidence for hyperactivation of platelets (healthy individuals, cancer patients receiving chemotherapy), which is likely why we did not observe evidence for platelet-derived cfDNA; yet, as mentioned, platelet-derived cfDNA may be an interesting analyte for studying hypercoagulable states, which may be an interesting avenue for future research. We also note two additional lines of evidence that platelets do not make appreciable contribution to cfDNA: 1) according to our estimations, platelets contain only 0.1% of the DNA of the megakaryocyte that generated them – making it more likely that the predominant source of megakaryocyte cfDNA is megakaryocytes; and 2) the surprising uneven genomic composition of platelets should have been represented in cfDNA if this was a significant source of cfDNA. The fact that cfDNA has a balanced representation of the genome further suggests that platelets are not significant contributors.

We have now further clarified this in the discussion of the article (line 288).